# Acute Effect of Night Shift Work on Endothelial Function with and without Naps: A Scoping Review

**DOI:** 10.3390/ijerph20196864

**Published:** 2023-09-29

**Authors:** Paul D. Patterson, Jacob C. Friedman, Samuel Ding, Rebekah S. Miller, Christian Martin-Gill, David Hostler, Thomas E. Platt

**Affiliations:** 1Department of Emergency Medicine, School of Medicine, University of Pittsburgh, Pittsburgh, PA 15261, USA; 2Department of Community Health Services and Rehabilitation Sciences, School of Health and Rehabilitation Sciences, University of Pittsburgh, Pittsburgh, PA 15261, USA; 3Health Sciences Library System, University of Pittsburgh, Pittsburgh, PA 15261, USA; 4Department of Exercise and Nutrition Sciences, School of Public Health and Health Professions, University at Buffalo, The State University of New York, Buffalo, NY 14214, USA; dhostler@buffalo.edu

**Keywords:** shift work, endothelial function, cardiovascular disease, napping, scoping literature review

## Abstract

We examined the breadth and depth of the current evidence investigating napping/sleeping during night shift work and its impact on non-invasive measures of endothelial function. We used a scoping review study design and searched five databases: Ovid Medline, EMBASE, Ovid APA PsycInfo, Web of Science Core Collection, and EBSCO CINAHL. We limited our search to English language and publications from January 1980 to September 2022. Our reporting adhered to the PRISMA-ScR guidance for scoping reviews. Our search strategy yielded 1949 records (titles and abstracts) after deduplication, of which 36 were retained for full-text review. Five articles were retained, describing three observational and two experimental research studies with a total sample of 110 individuals, which examined the non-invasive indicators of endothelial function in relation to the exposure to night shift work. While there is some evidence of an effect of night shift work on the non-invasive indicators of endothelial function, this evidence is incomplete, limited to a small samples of shift workers, and is mostly restricted to one measurement technique for assessing endothelial function with diverse protocols. In addition, there is no identifiable research investigating the potential benefits of napping during night shift work on non-invasive measures of endothelial function.

## 1. Introduction

Cardiovascular disease (CVD) is the leading cause of death in the U.S. and globally [1]. The risk of CVD and the incidence of hypertension over a 5- to 10-year period is higher among shift workers compared to traditional daylight workers [2,3]. A shift work schedule, compared to a non-shift work daytime schedule, is associated with a 23% increased risk of myocardial infarction (risk ratio 1.23, 95%CI 1.15, 1.31) and a 5% increased risk of ischemic stroke (risk ratio 1.05, 95%CI 1.01, 1.09) [4,5]. The risk of myocardial infarction, coronary-related mortality, or hospital admission due to coronary artery disease is greatest among night shift workers compared to other shift schedules (risk ratio 1.41, 95%CI 1.13, 1.76) [5]. Organizations such as the U.S. National Institutes for Occupational Safety and Health (NIOSH), the U.S. Federal Emergency Management Agency, the World Health Organization, and the International Labour Organization (ILO) support the risk reduction in CVD among shift workers [6,7,8]. Common risk mitigation strategies include the frequent assessment of non-invasive indicators of CVD, such as blood pressure (BP) [9]. The monitoring of other pre-clinical indicators, such as endothelial function, may provide greater insight into CVD risk and disease progression among shift workers [10,11,12,13], yet little is known about the evidence for the indicators of endothelial function, including sensitivity to on-shift interventions such as napping.

The endothelium is a single-layer of cells that line the vascular system, including arteries, veins, and capillaries, and it is an important modulator of vascular tone, BP, and other critical functions that impact cardiovascular homeostasis [14]. The endothelium plays an important role in coagulation and clotting processes, which has an impact on the development of atherosclerosis [15]. Endothelial dysfunction can manifest as changes in molecule secretion, inflammation in the vessel wall, and as increased BP [11]. Prior studies report circadian variation in endothelial function marked by the attenuation of function in the early morning compared to evening [16]. Long-term indicators of endothelial dysfunction include arterial stiffness and evidence of atherosclerosis [15].

Advantages of assessing endothelial function, especially non-invasively, include (1) less risk compared to invasive techniques like the intra-coronary infusion of acetylcholine to measure arterial diameter [17]; (2) mostly all CVD risk factors are linked to endothelial dysfunction [15,18]; (3) because endothelial dysfunction is systemic, dysfunction detected in the peripheral vasculature correlates with dysfunction in the coronary arteries [19]; (4) numerous studies using non-invasive techniques report statistically significant associations with CVD outcomes [18,20,21]; and (5) the best practice for monitoring other indicators, like BP, often require prolonged 24 h monitoring for diagnostic purposes [22], which may not be feasible in all workplace wellness and shift work settings.

Shift work is described as working hours outside of what may be defined as a normal working day (e.g., 9 a.m. to 5 p.m.) [23]. Night shift work encompasses work beyond midnight for 3 or more hours [23]. According to one estimate, 16.8% of all full-time wage and salary workers in the United States (U.S.) are shift workers [24]. Other estimates suggest that one-fifth of workers in the U.S. and European Union (E.U.) are shift workers, with as much as one-third of employees in the U.S. and E.U. working non-standard hours and/or >48 h per week [25]. Shift work schedules such as night shifts, long duration shifts, and rotating shift work are linked to shortened sleep (by 1–2 h compared to non-shift workers), disrupted sleep, and poor sleep quality [23]. Sleep loss associated with shift work, especially night shift work, has been shown to disrupt normal patterns in BP and affect the normal functions of the endothelium [13,26,27]. The repeated disruptions of BP and endothelial function associated with the frequent exposure to shift work may exacerbate the risk of CVD [10,28].

The studies of acute (e.g., days) and longer term (e.g., years) exposure to shift work show an association between shift work and endothelial dysfunction [10,13,28,29]. Suessenbacher and colleagues reported reduced peripheral arterial tone—a non-invasive measure of endothelial function—among shift workers with a mean of 9.9 years (SD5.3) of work experience when compared to non-shift workers with a mean of 14.5 years (SD8.9) of work experience [10]. A separate study of nurses showed a significant decrease in endothelial function after three consecutive night shifts when compared to pre-night shift work (baseline) [29]. In this study, a longer history of night shift work was strongly associated with a decrease in endothelial function, as measured via flow-mediated dilation (FMD) [29]. Charles and colleagues compared measures of FMD assessed 7 years apart in police officers and determined that officers who worked night shifts experienced greater declines in endothelial function than officers who worked daylight or afternoon shifts [13].

Napping during night shifts briefly restores normal (wake/sleep) patterns in BP [27,30], which may directly or indirectly impact the relationship between sleep loss during night shift work and endothelial function [31]. A normal pattern of BP is marked by a 10–20% decrease (dip) in BP during sleep/nighttime hours relative to the wake/daytime hours [32]. Napping refers to “*sleep periods at least 50% shorter than an individual’s average nocturnal sleep length*” [33,34]. A study of 56 Emergency Medical Services (EMS) night shift workers showed that a nap of 60 min or longer during night shift work was associated with the restoration of normal BP dipping, for most, during the night shift work hours [27]. A recent meta-analysis of 24 h shifts showed that Systolic BP (SBP) and Diastolic BP (DBP) were significantly higher during wakefulness on the long duration shift compared to sleep during this shift [30]. This meta-analysis showed that among those who slept during the 24 h shift, the pooled mean dip in SBP was 14.8% (95%CI 11.4, 18.2) and the mean dip in DBP was 17.1% (95%CI 13.6, 20.6) [30]. Evidence of a napping effect on endothelial function could impact how and when it is assessed in relation to the exposure to shift work.

We sought to examine the breadth and depth of the current evidence investigating napping/sleeping during night shift work and its impact on non-invasive measures of endothelial function. Our findings will inform others guided by the work of the ILO, NIOSH, and other organizations (public and private) who regularly monitor and investigate the working conditions and/or offer guidance or resources for employers responsible for mitigating threats to shift worker health and safety [25,35,36]. Our findings will inform researchers who address NIOSH’s Strategic Goals, Intermediate Goals, and Activity Goals for occupational health and safety, specifically Intermediate Goals 1.10, 1.11, 1.13, and 7.17, and 7.1–7.12, 7.14, 7.5, 7.6, and 7.10 of the NIOSH Strategic Plan for 2019–2026, and also the corresponding Activity Goals, which target work arrangement and scheduling as risk factors for illnesses like CVD [36]. The findings from our review will inform researchers as well as decision-makers who seek to address the ILO’s policy suggestions, such as “Developing balanced working time arrangements” that promote health and safety [25]. In addition, our review will inform those following or taking action on NIOSH’s Future of Work Initiative, which promotes collaboration and new research to address workplace-related risks created by job arrangements and organizational design [35]. The Initiative also promotes creating risk profiles and new approaches to mitigate risk with “new solutions and practical approaches” [35]. Irrespective of the goal, objective, initiative, or policy suggestion selected, researchers and individuals responsible for policy-relevant decisions should be cognizant of the best available evidence (“be aware of what is out there”) for measuring and quantifying the indicators of risks (or for assessing intervention impact) with respect to a key pre-clinical indicator of CVD. In this review, we will identify and characterize the breadth and depth of evidence related to a potentially useful and clinically meaningful approach to assessing risk (or intervention impact) for CVD.

## 2. Materials and Methods

### 2.1. Study Design

Our review was initially guided by a single research question framed in the traditional Population, Intervention/Exposure, Comparison, Outcome (P.I.C.O.) format: “*Does napping/sleeping during night shift work mitigate the impact of sleep deprivation/sleep restriction on indicators of endothelial function*?” Following a systematic search of the literature identifying no studies directly answering this question, we reframed our search as a scoping review of the literature aimed at investigating the breadth and depth of the current evidence evaluating the relationship between napping/sleeping during night shift work and endothelial function. We used a scoping review study design given its utility for investigations focused on the depth and breadth of the evidence, the need to map gaps in the evidence, and because of the uncertainty regarding the potential number of original research studies that addressed our P.I.C.O. [37].

### 2.2. Search Strategy

As prescribed [37], we conducted a comprehensive search of multiple databases (*n* = 5) for original research relevant to our original P.I.C.O. research question: Ovid Medline, EMBASE, Ovid APA PsycInfo, Web of Science Core Collection, and EBSCO CINAHL. For each database search, we used controlled vocabulary and keywords for the concepts of endothelial function and shift work, and we limited our search date range from 1 January 1980 to 8 September 2022. The search results were deduplicated using the Amsterdam Efficient Deduplication method [38] and then uploaded into Endnote citation management software. The records were then deduplicated a second time using the Bramer method [39]. See Appendix A for details of our search strategy stratified by database. The records resulting from deduplication were then uploaded to DistillerSR (DistillerSR Inc., Ottawa, ON, Canada).

### 2.3. Screening Methodology

The adjudication of records was guided by a priori inclusion/exclusion criteria applied by two independent record reviewers. Agreement between reviewers during the record screening was assessed with the Kappa statistic. As prescribed [37], the records retained during the title/abstract screening were examined in full-text form, adjudicated for inclusion or exclusion, and the data from the included full-text articles were extracted into tables and verified by two investigators (See Appendix B). The extracted data were descriptively synthesized, where the reasons for excluding articles appear in Appendix C, and the presentation of results adhered to the PRISMA-ScR flow diagram (See Appendix D) [40]. We also performed bibliography searches during the review of full-text articles to identify potentially relevant research not identified during the record screening. Our protocol was not published or registered in advance.

### 2.4. Population of Interest

Our population of interest was shift workers from any occupation (e.g., public safety, healthcare, transportation, manufacturing, or related shift worker groups). Original research studies that involved military personnel as study participants were included.

### 2.5. Intervention/Exposure of Interest

The intervention or exposure of interest was napping during night shift work, long duration shifts (e.g., 24 h), or during simulated night shifts or long duration shift work.

### 2.6. Comparison of Interest

The comparison of interest was on the impact of napping/sleeping versus no napping/sleeping during night shift work on endothelial function.

### 2.7. Outcomes of Interest

Our primary outcome of interest was the indicators of endothelial function or dysfunction measured with non-invasive techniques. Non-invasive assessments include brachial arterial reactivity tests that involve ultrasound, venous occlusive plethysmography that involve BP cuffs placed on the lower and/or upper extremities, and peripheral tonometry, which involves placing probes on a subject’s fingers to assess microvasculature blood flow before, during, and after occlusion of the upper extremity [14,41]. The gold standard measurement of endothelial function involves the invasive assessment of arterial diameter, blood flow, and vascular resistance with intra-coronary infusion of acetylcholine [17]. Biomarkers such as IL-6, C-reactive protein (CRP), and syndecan-1, among others, are indicators of endothelial activation or dysregulation [42,43]. We focused on non-invasive assessments of endothelial function given greater feasibility, compared to invasive methods, of replicating the use of non-invasive devices in future research with diverse shift worker populations.

### 2.8. Analysis

As prescribed [37,40], we used a narrative and descriptive approach to analyze the retained literature. We approached the analyses of the retained literature in this way given that previous research reports that two-thirds of the literature screened for studies that involve systematic reviews and other types of reviews often do not report on the comparison of interest [44]. In this scoping review, our comparison of interest focused on the impact of napping/sleeping during night shifts versus no napping/sleeping and the effect on endothelial function. While we anticipated that a few of the articles reviewed would address all aspects of our P.I.C.O., we took particular interest in the articles (studies) that met most of our P.I.C.O. elements. We closely examined the methods of these articles and reported the findings for the purposes of highlighting important gaps in the evidence. We refer to the articles (studies) that met most of the elements of our P.I.C.O., but not all, as ‘ancillary.’ Highlighting such gaps may offer much needed guidance for future studies. In this analysis, and as recommended by others [37,40], we charted and outlined key findings of all the retained articles in evidence tables (see Appendix B), and below, in the Results and Discussion sections, we use a narrative format to describe important methodological gaps in these studies for the benefit of future research.

## 3. Results

Our search strategy yielded 1949 records (titles and abstracts) after deduplication, of which 36 were retained for full-text review (Figure 1). Inter-rater agreement at the screening phase was moderate (Kappa = 0.43) and comparable to the agreement reported in previous reviews [45,46]. We searched bibliographies of all 36 retained publications and identified an additional four potentially relevant articles. In total, 40 full-text articles were evaluated against our P.I.C.O. criteria. Five articles were retained, describing three observational and two experimental research studies that targeted shift workers and examined non-invasive indicators of endothelial function in relation to the exposure to night shift work. However, none of these five studies addressed the comparison of interest: the effect of napping/sleeping during night shift work (compared to no napping/sleeping) on the non-invasive indicators of endothelial function. We charted the key findings from these five ancillary articles [28,47,48,49,50], which can be accessed in Appendix B. Below, we describe these studies along with the research gaps. In total, we excluded 35 articles out of the 40 assessed during the full-text review that did not target shift workers or failed to address multiple elements of our P.I.C.O. The reasons given for exclusion appear in Appendix C. All the publications reviewed as part of the bibliography search were excluded.

### 3.1. Description of Populations Studied

We identified five unique ancillary studies with a cumulative study sample of n = 110 individuals, of which 87% were shift workers [28,47,48,49,50]. Amir and colleagues enrolled 30 healthy physicians working at a hospital in Israel [28]. Among the enrolled, 22 were physician residents (17 internal medicine and five surgery) and eight fellows (six cardiology, one gastroenterology, and one hematology). Garu and colleagues enrolled 13 hospital workers with night duty working at a hospital in Japan [47]. This group included six men and seven women. Tarzia and colleagues recruited 20 cardiology trainees from a university in Italy [48]. This group included nine males and 11 females. Wehrens and associates enrolled 11 male shift workers and 14 non-shift workers between the ages of 25 and 45 years in the United Kingdom [49]. Zheng and colleagues enrolled 22 internal medicine residents in the U.S., of which seven were women [50]. The participant demographics reported for all the retained studies describe the participants as generally healthy and absent of medical conditions (e.g., hypertension) that may impact endothelial function following the exposure to night shift work.

### 3.2. Description of Study Design and Exposure

Three studies used a prospective observational study design [28,47,48], while two used a randomized crossover study design [49,50]. In the observational designs [28,47,48], investigators obtained the baseline measures of endothelial function prior to a standard daylight shift. The follow-up measures of endothelial function were obtained after the exposure to night shift work, most often in the morning hours post-night shift. In the one laboratory-based study, Zheng and colleagues required participants to abstain from caffeine during the study protocol and then performed two assessments of endothelial function in random order [50]. One assessment was performed after 1 p.m., following a 30 h extended shift, and the second was obtained after 1 p.m., following a 6 h daylight shift (7 a.m.–1 p.m.).

Our intervention/exposure of interest was napping/sleeping during night shift work. One study measured sleep with a questionnaire completed after night shift work [28]. Zheng and colleagues required subjects record their sleep hours during shift work on a paper-based diary and return these diaries during the post-shift assessments of endothelial function [50]. Two studies provided little or no detailed description of the methods used to document sleep during night shift work [47,48]. One laboratory-based study monitored sleep with actigraphy and polysomnography [49].

The study by Amir and colleagues reported 50% (n = 15) of participants obtaining ≥3 h of sleep during a 24 h shift and another 50% (n = 15) of participants obtaining < 3 h of sleep during the shift [28]. In the study by Garu and associates, the participants reported a mean of 2.3 h (SD1.0) of sleep during nighttime work (notably, the hours-of-work duration is not reported) [47]. Tarzia and colleagues reported 55% of participants (n = 11) obtaining <4 h of sleep during night shift work and 45% (n = 9) obtaining >4 h of sleep [48]. Zheng and colleagues reported participants obtaining an average of 0.3 h of sleep (IQR 0.0–1.5) during an extended shift of 30 h [50]. In these studies, the timing of sleep was not reported. In the laboratory-based study by Wehrens and colleagues, the participants were kept in the lab for 4 consecutive days that included a 30.5 h period of sleep deprivation [49]. The protocol allowed participants to sleep before and after the extended period of wakefulness, but not during.

### 3.3. Description of Comparisons

None of the retained studies included the comparison of interest, which was napping or sleeping during night shift work (intervention) versus no napping or no sleep during night shift work (comparison).

### 3.4. Description of Outcome Measures

Four studies used non-invasive FMD to measure endothelial function [28,48,49,50], whereas one study used the EndoPAT device (ZOLL^®^ Intamar^®^, Atlanta, GA, USA) [47]. The lower levels of FMD, expressed as a percentage, suggest impaired endothelial function. Of the four studies using FMD, one performed the FMD measurements at 1 p.m., after a traditional daylight shift, and again at 1 p.m., after a 30 h extended shift [50]. Authors reported impaired FMD following the extended shift when compared to the measures after the daylight shift (*p* < 0.05) [50]. Two studies performed two different FMD measurements in the morning (e.g., 8–9/10 a.m.), with one occurring after a night of normal sleep and again after a night shift [28,48]. Both studies showed lower FMD after a night shift compared to FMD assessed after a night of normal sleep (*p* < 0.05) [28,48]. One study performed two FMD measurements per day (once in the morning and once in evening) during a 4-day long laboratory protocol [49]. Investigators focused their analyses on the differences between shift workers and non-shift workers and detected no differences between these groups for any of the measures [49].

Two studies performed a single FMD measurement per examination session (e.g., once after a night shift) [28,50], whereas two used repeated measures [48,49]. In addition to the standard FMD measurements, two studies involved additional Nitrate-Mediated Dilation (NMD) measurements, with one study using a 25 mcg sublingual glyceryl trinitrate dose protocol [48], and one study using a 400 mcg glyceryl trinitrate sublingual spray [28]. Two studies that used FMD reported applying the ultrasound probe to the antecubital fossa and using a reactive hyperemia protocol with a brachial-placed BP cuff inflated to 200–300 mmHg for 5 min or longer [48,49]. Amir and colleagues positioned the BP cuff on the forearm [28]. Three studies reported that prior to performing the FMD measurements, the participants were placed in a supine position following a 5 to 10 min rest [28,48,49]. Three studies reported that expert or highly qualified/experienced operators performed the FMD measurements [28,48,49]. Zheng and colleagues did not report the details of their FMD protocol [50].

Garu and colleagues used the EndoPAT device for the assessment of endothelial function [47]. The EndoPAT device evaluates plethysmographic changes in micro-circulation (arterial pulse pressure/tonometry), often in the index finger before, during, and after a brief period (i.e., 5 min) of occlusion with a brachial-positioned BP cuff. The EndoPAT reported outcome is the Reactive Hyperemia Index (RHI). Previous research suggests RHI values lower than 1.67 as abnormal or are a pre-clinical indicator of endothelial dysfunction [51]. Garu and colleagues reported that study participants were assessed with the EndoPAT a total of six times (three measures before a daylight shift and three measures after a night shift) [47]. All participants were placed in a seated position with their hands at heart level during the measurements, the probes were placed on the index finger, and the BP cuff was placed on the upper arm and inflated to at least 200 mmHg during the occlusion phase [47]. The reported findings show no differences between the pre-daylight shift and post-night shift measures (*p* > 0.05) [47].

## 4. Discussion

The major findings from this scoping review include (1) previous research which suggests that the exposure to night shift work has an acute (post-night shift), negative impact on endothelial function; (2) among the studies reviewed, most involved small samples of shift workers and widely variable protocols; and (3) there is an absence of research investigating the impact of napping during night shifts on the non-invasive measures of endothelial function.

The implications of our findings for researchers and individuals concerned with or seeking to operationalize the ILO’s “Future of Work” policy suggestions and/or NIOSH’s Future of Work Initiative, or NIOSH’s Strategic Plan, is two-fold. First, researchers may use our findings to decide whether the current evidence is adequate to support the use of non-invasive assessments of endothelial function when investigating work scheduling (organizational design) and its impact on the risk of CVD. Second, researchers and policy makers must decide whether it is appropriate to invest in new/novel measurement tools and techniques for future use, like the non-invasive EndoPAT device, because these devices and approaches have the potential to assess and quantify risks at an earlier stage in disease progression compared to the current indicators like BP. How researchers and policy makers respond to these questions will have a direct impact on the attention given to the ILO’s “Future of Work” efforts and NIOSH’s Future of Work Initiative and Strategic Plan [25,35,36].

While a sizeable gap in research is identified, our scoping review uncovers opportunities for future investigations that could shed much needed light on the impact that night shift work has on the key indicators of shift worker cardiovascular health and if evidence-based interventions like napping have a clinically meaningful effect. A specific opportunity for future research is, in an adequately powered study, the comparison of napping or sleeping during night shift work versus no napping or no sleeping. This or related studies would test interventions that could be implemented in shift work occupations to improve occupational health and safety. Investigators pursuing such opportunities would directly address numerous goals outlined by leading workplace health and safety organizations like NIOSH (e.g., Intermediate Goals 1.10, 1.11, 1.13, and 7.17, and 7.1–7.12, 7.14, 7.5, 7.6, and 7.10 of the NIOSH Strategic Plan for 2019–2026) [35,36].

Numerous epidemiological studies show a consistent and compelling relationship between the years of exposure to shift work and CVD risk [2,3,4,5]. Interest among researchers in monitoring endothelial function in relation to CVD risk is substantial and will likely grow because dysfunction can be detected non-invasively and before clinical symptoms of CVD manifest and require intervention (i.e., hypertension) [14,15]. In short, monitoring endothelial function in relation to the exposure to night shift work gives researchers and clinicians targets for intervention and opportunities to assess intervention efficacy and effectiveness [13,14,29]. The findings from our scoping review show that while there is likely great interest and promise in monitoring endothelial function in relation to night shift work, the depth and breadth of existing research is far from comprehensive. Recent research demonstrates the benefits of napping during night shifts on the key indicators of CVD risk (i.e., BP) [27,30,52]. However, in this scoping review, we failed to identify a similar line of research exploring the potential benefits of night shift naps on endothelial function. While surprising, our findings suggest this is a prime area for further investigation.

Additional studies exploring the impact of night shifts on endothelial function, and the potential beneficial effects of napping (for example), could benefit from adopting rigorous experimental study designs. Most of the studies highlighted in this review were observational and the vast majority relied on one approach to measuring endothelial function (i.e., Flow Mediated Dilation). The reproducibility of FMD measures has been questioned [41]. The test results are highly dependent on technician skill and interrater agreement can be low [18,41,53,54]. Other non-invasive tools exist (i.e., the EndoPAT device) and offer opportunities to limit the need for highly trained technicians. The EndoPAT device is innovative, yet its utility has been questioned given the findings in previous research showing a low correlation with FMD [55]. Regardless of the measurement device used, investigators can improve the quality of future research by using adequately powered studies and experimental designs such as within-subject randomized crossover trials. Crossover designs are appropriate for testing the effect of interventions like napping during night shifts because the response of interest in each condition is often proximate to the exposure, where the exposure or intervention of interest does not permanently alter the study outcomes and the power is maximized with the within-subject comparisons [56].

Future studies that target and enroll shift workers will build on existing research, which make a meaningful contribution to the literature and provide stakeholders with coveted direct evidence. The studies of non-shift workers are informative, yet these data are indirect and are often downgraded in terms of the quality of evidence when collated with other studies [57]. These data are downgraded because targeted populations (i.e., shift workers) can differ from non-shift workers in meaningful ways, such as in chronotype [58]. These differences may have a clinically meaningful impact on the outcomes of interest (i.e., sleep duration) [59]. Thus, the studies focused on shift workers or shift work schedules stand a better chance of adding to the literature when enrolling shift workers as study subjects than studies with alternative populations.

### Strengths and Limitations

Common limitations of scoping reviews include incomplete database searches, the exclusion of research librarians designing search strategies, incomplete descriptions of the search and screening methods, limited details on the information abstracted from the retained literature, and variable methods for the synthesis of results [37,40]. We addressed these limitations in advance by (1) searching multiple databases and outlining the details of our search strategies in Appendix A; (2) our team included an experienced medical research librarian (RSM) with expertise in developing and refining search strategies; and (3) as prescribed [40], we documented the key findings in tables and provided a detailed, narrative, and descriptive summary of the results. We strengthened our methodology by searching the bibliographies of the retained literature to identify potentially relevant research. Finally, we adhered to recent guidance from the Preferred Reporting Items for the Systematic Reviews and Meta-Analyses extension for Scoping Reviews to ensure transparency and the opportunity for others to replicate our approach [40]. Despite these strengths, our study is limited like other reviews are limited, by (1) excluding the gray (unpublished) literature, and (2) the judgment of our investigators at multiple phases of our review. Specifically, our decisions and judgment to include or exclude any literature may differ from others; therefore, our findings may differ from other investigators.

## 5. Conclusions

While there is some evidence of an effect of night shift work on non-invasive indicators of endothelial function, this evidence is incomplete, limited to small samples of shift workers, and is mostly restricted to one measurement technique. In addition, there is no identifiable research investigating the potential benefits of napping during night shift work on the non-invasive measures of endothelial function. The limitations identified in this scoping review are also opportunities for future research that may have a meaningful impact on shift worker health and wellbeing if adequately powered and inclusive of shift workers as study subjects—providing much needed rigor in the study design and direct evidence to those who employ shift workers.

## Figures and Tables

**Figure 1 ijerph-20-06864-f001:**
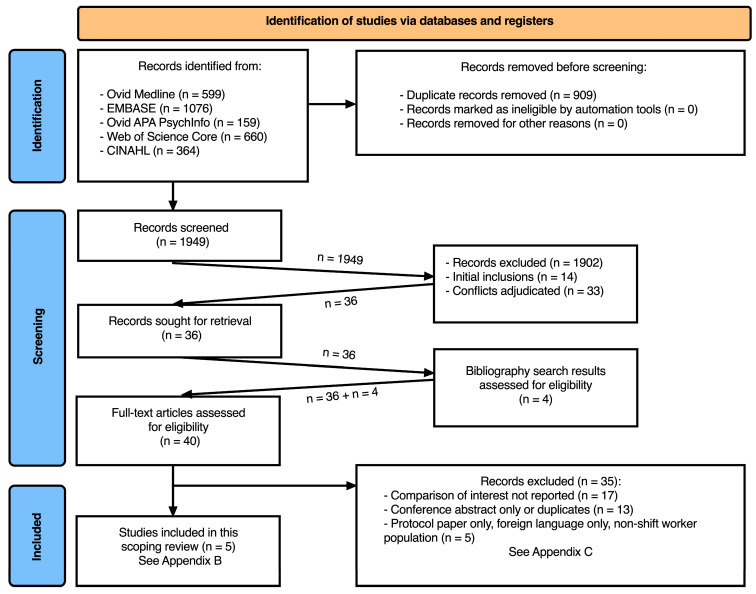
PRISMA flow diagram.

## Data Availability

Requests of the data reviewed in full-text format for this scoping literature review may be sent to the corresponding author.

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
