# Peer review of "Acute Effect of Night Shift Work on Endothelial Function with and without Naps: A Scoping Review"

_ijerph, 2023, doi:10.3390/ijerph20196864_

Round 1

Reviewer 1 Report

This is a well-written, interesting scoping review highlighting a need to advance science for shift workers. Here are a few comments to improve the manuscript: 

Under the analysis section the term "comparison of interest" is not clearly stated - it is described fully in results but should be defined on first mention. Authors should add what this refers to specifically for clarity. In general, this paragraph should be reviewed for clarity and accuracy of the actual analysis process so that it could be replicated if desired.  

Description of population would benefit by describing who those 13% "not shift workers" were. 

Description of comparison on interest in results should summarize that it was not present at all in the selected studies if this was so.

Author Response

REVIEWER 1:

REVIEWER 1: This is a well-written, interesting scoping review highlighting a need to advance science for shift workers.

AUTHOR RESPONSE:  Sincerely thank you for the comment.

REVIEWER 1: Here are a few comments to improve the manuscript: 

Under the analysis section the term "comparison of interest" is not clearly stated - it is described fully in results but should be defined on first mention. Authors should add what this refers to specifically for clarity. In general, this paragraph should be reviewed for clarity and accuracy of the actual analysis process so that it could be replicated if desired.  

AUTHOR RESPONSE:  Thank you for this comment and request for clarification. We have edited the Analysis section and pasted the updated/edited text below. We hope this edit improves clarity and addresses your comment.

“As prescribed[34, 37], we used a narrative and descriptive approach to analyze retained literature. We approached analyses of retained literature in this way given that previous research reports two-thirds of the literature screened for studies that involve systematic reviews, and other types of reviews, often do not report on the comparison of interest [41]. In this scoping review, our comparison of interest focused on the impact of napping/sleeping during night shifts versus no napping/sleeping and the effect on endothelial function. While we anticipated few of the articles reviewed would address all aspects of our P.I.C.O., we took particular interest in articles (studies) that met most of our P.I.C.O. elements. We closely examined the methods of these articles and reported findings for purposes of highlighting important gaps in evidence. We refer to articles (studies) that met most elements of our P.I.C.O., but not all, as ‘ancillary.’ Highlighting such gaps may offer much needed guidance for future studies. In this analysis, and as recommended by others[34, 37], we charted and outlined key findings of all retained articles in evidence tables (see Appendices) and below in the Results and Discussion sections we use a narrative format to describe important methodological gaps in these studies for the benefit of future research.”

REVIEWER 1: Description of population would benefit by describing who those 13% "not shift workers" were. 

AUTHOR RESPONSE:  The non-shift workers originate from the Wehrens et al., 2012 study (PMID-21953310). Because the authors provide little information about these non-shift workers, we are unable to provide much more than characterizing them as non-shift workers in our description. We agree that having more information about these subjects would provide perspective and context.

REVIEWER 1: Description of comparison on interest in results should summarize that it was not present at all in the selected studies if this was so.

AUTHOR RESPONSE:  We agree with the reviewer’s comment and have edited the statements in this sub-section to now read as: “None of the retained studies included the comparison of interest, which was napping or sleeping during night shift work (intervention) versus no-napping or no sleep during night shift work (comparison).” We hope this clarifies.

Reviewer 2 Report

The manuscript concerns the current evidence investigating napping/sleeping during night shift work and its impact on non-invasive measures of endothelial function. In short, there is limited evidence of an effect of night shift work on non-invasive indicators of endothelial function, and there is no identifiable research investigating the potential benefits of napping during night shift work on non-invasive measures of endothelial function.

This scoping review is expected to provide valuable insights for designing additional research regarding the effects of night shift work on non-invasive measures of endothelial function, as well as the potential beneficial role of napping during night shifts. I have just a few minor recommendations:

1.     [Introduction] Please provide a more detailed explanation of the advantages of measuring non-invasive endothelial function compared to other non-invasive indicators, such as blood pressure, in the management of CVD risk. In addition, please offer information regarding the correlation between non-invasive measures of endothelial function, such as FMD, and CVD risk. This will enable researchers and policymakers better comprehend the value of using non-invasive endothelial function in monitoring the risk of CVD in shift workers.

2.     [Figure 1] Please explain what “*” and “**” signifies.

3.     [Figure 1] Please specify the number of inclusions or exclusions for each direction in the bidirectional arrows.

4.     [Discussion] In the Limitations section, many strengths of this paper are introduced. Therefore, I suggest titling this section 'Strengths and Limitations'.

Author Response

REVIEWER 2:

REVIEWER 2: The manuscript concerns the current evidence investigating napping/sleeping during night shift work and its impact on non-invasive measures of endothelial function. In short, there is limited evidence of an effect of night shift work on non-invasive indicators of endothelial function, and there is no identifiable research investigating the potential benefits of napping during night shift work on non-invasive measures of endothelial function. This scoping review is expected to provide valuable insights for designing additional research regarding the effects of night shift work on non-invasive measures of endothelial function, as well as the potential beneficial role of napping during night shifts. I have just a few minor recommendations:

AUTHOR RESPONSE:  On behalf of all authors, thank you for the positive comments.

REVIEWER 2:   [Introduction] Please provide a more detailed explanation of the advantages of measuring non-invasive endothelial function compared to other non-invasive indicators, such as blood pressure, in the management of CVD risk. In addition, please offer information regarding the correlation between non-invasive measures of endothelial function, such as FMD, and CVD risk. This will enable researchers and policymakers better comprehend the value of using non-invasive endothelial function in monitoring the risk of CVD in shift workers.

AUTHOR RESPONSE:  We thank the reviewer for this comment. We have added the following paragraph to the Introduction section, which we hope addresses the reviewer’s concerns.

“Advantages of assessing endothelial function, especially non-invasively, include: (1) less risk compared to invasive techniques like intra-coronary infusion of acetylcholine to measure arterial diameter [17]; (2) most all CVD risk factors are linked to endothelial dysfunction [15, 18]; (3) because endothelial dysfunction is systemic, dysfunction detected in the peripheral vasculature correlates with dysfunction in the coronary arteries [19]; (4) numerous studies using non-invasive techniques report statistically significant associations with CVD outcomes [18, 20, 21]; and (5) best practice for monitoring other indicators, like BP, often require prolonged 24-hour monitoring for diagnostic purposes[22], which may not be feasible in all workplace wellness and shift work settings.”

REVIEWER 2: [Figure 1] Please explain what “*” and “**” signifies.

AUTHOR RESPONSE:  We apologize for what appears to be a typo error in the images. These have been removed.

REVIEWER 2:  [Figure 1] Please specify the number of inclusions or exclusions for each direction in the bidirectional arrows.

AUTHOR RESPONSE:  We have edited the figure as requested.

REVIEWER 2: [Discussion] In the Limitations section, many strengths of this paper are introduced. Therefore, I suggest titling this section 'Strengths and Limitations'.

AUTHOR RESPONSE:  We have edited as requested but defer to the Journal for their preferred style.